



# Free-vortex models for wind turbine wakes under yaw misalignment – a validation study on far-wake effects

Maarten J. van den Broek[1], Delphine De Tavernier[2], Paul Hulsman[3], Daan van der Hoek[1], Benjamin Sanderse[4], and Jan-Willem van Wingerden[1]

[1]Delft Centre for Systems and Control, TU Delft, Mekelweg 2, 2628CD Delft, NL
[2]Department of Flow Physics and Technology, Wind Energy, TU Delft, Kluyverweg 1, 2629HS Delft, NL
[3]ForWind – Institute of Physics, University of Oldenburg, Küpkersweg 70, 26129 Oldenburg, DE
[4]Scientific Computing, CWI, P.O. Box 94079, 1090GB Amsterdam, NL

**Correspondence:** Maarten J. van den Broek (m.j.vandenbroek@tudelft.nl)

**Abstract.** Near-wake effects of wind turbine models using the free-vortex wake have been extensively studied, but there is a lack of validation for such predictions in the mid to far wake. This paper presents a novel validation study using three free-vortex wake models of increasing complexity: an actuator disc, an actuator disc with rotation, and an actuator-line model. We emphasise the application for dynamic wind farm flow control optimisation with a focus on wake redirection using yaw misalignment. For that purpose, surrogate wake models should provide sufficiently accurate power predictions at low computational expense. Three sets of wind tunnel data are used for validation: flow measurements under steady yaw misalignment, time-resolved flow measurements for a step change in yaw, and turbine output measurements with yaw control and simulated wind direction variation. Results indicate that the actuator-disc model provides the best balance of computational cost and accuracy in power predictions for the mid to far wake, which is not significantly improved upon by the addition of rotation. In the near wake, the added complexity of the actuator-line model may provide value as it models blade loading and individual tip vortices. Altogether, this study conclusively demonstrates that the actuator-disc model is suitable for yaw control optimisation and provides important validation for further studies into optimisation of wake steering under time-varying conditions.

## 1 Introduction

The limited availability of offshore and onshore parcels for wind energy production means that large, densely-spaced wind farms are commonly used. However, in these farms, wake effects can lead to a significant decrease in power production and an increase in fatigue loading. While farm topology is typically optimised to minimise aerodynamic interaction, it lacks flexibility for time-varying wind conditions (van Wingerden et al., 2020). Under those conditions, wind farm flow control uses existing control degrees-of-freedom to reduce aerodynamic interaction between wind turbines with methods such as wake redirection through yaw misalignment, dynamic induction control with collective pitch control, and wake mixing strategies with individual pitch control (Meyers et al., 2022).

This paper focuses on the use of yaw misalignment for wake steering, where an intentional misalignment in the yaw angle with respect to the dominant wind direction is used to deflect the low energy, turbulent wake behind the turbine. After





demonstrations of effectiveness in both simulation and wind tunnel experiments, wake steering has been shown to yield power gains in wind farms for pre-defined yaw-angle offsets under steady conditions in field studies (Howland et al., 2019; Fleming et al., 2020, 2021; Doekemeijer et al., 2021; Simley et al., 2021). An important aspect of the wind turbine wake under yaw misalignment is the formation of a counter-rotating vortex pair which generates a curled, or kidney-shaped, wake (Howland et al., 2016; Bastankhah and Porté-Agel, 2016; Bartl et al., 2018; Fleming et al., 2018; Hulsman et al., 2022b).

Wind turbine wake models are essential tools for developing and implementing wake steering control strategies. Accurate predictions of wake behaviour allow optimisation of wind turbine controls for objectives such as power production and reduction of fatigue loading. Current control strategies are mostly based on look-up tables generated by steady-state optimisation with engineering wake models, such as those in the FLORIS toolbox (NREL, 2022). This includes for example the curled wake model (Martínez-Tossas et al., 2019), which has been extended with dynamics (Branlard et al., 2023) as the steady-state models are limited for use in time-varying conditions. Another approach is the use of Lagrangian particle methods to use the wakemodels within FLORIS for dynamic wake prediction (Becker et al., 2022; Lejeune et al., 2022).

Instead of implementing the dynamics into steady-state models, physics-based approaches attempt to simplify first principles to reduce complexity while maintaining essential dynamics. Studies with large-eddy simulation have been successful in control optimisation, but remain far from real-time application because of the associated computational cost (Munters and Meyers, 2018). On the other hand, simplified flow models based on two-dimensional Navier-Stokes equations, such as WFSim (Boersma et al., 2018) and FRED (van den Broek and van Wingerden, 2020), were developed in an attempt to provide computationally efficient flow estimates for control. However, these have been shown to be unsuitable for yaw control as the wake dynamics under yaw misalignment are fundamentally three-dimensional (van den Broek et al., 2022b). A physics-based model for efficient control optimisation was introduced by van den Broek et al. (2022a), modelling the wind turbine wake with an actuator-disc model based on free-vortex methods and representing the curled wake dynamics.

Free-vortex wake (FVW) methods are meshless methods, using Lagrangian elements to model flow dynamics based on the vorticity formulation of the Navier-Stokes equations (Leishman, 2000; Katz and Plotkin, 2001). This leads to an efficient, skeletal representation of the wind turbine wake. The FVW method has been initially applied to wake modelling for helicopter rotors with a focus on tip vortices in studies (Bhagwat and Leishman, 2001). From there, it was adapted to application for unsteady aerodynamics in wakes of wind turbine rotors (Leishman, 2002).

Several studies have applied FVW methods to model wind turbine wakes, such as a lattice method (Simoes and Graham, 1992), axisymmetric vortex rings for the near wake under yaw misalignment (De Vaal et al., 2014), and the study of the effectiveness of dynamic induction control for near-wake breakdown (Brown et al., 2021). The latter model has been shown to only be accurate within the near wake in a comparison with data from large-eddy simulations (Houck et al., 2022). Other studies utilise the flexibility of the meshless formulation for the study of wind turbines on floating platforms. These consider, for example, the effects of platform motion on rotor induction (Sebastian and Lackner, 2012), unsteady aerodynamics in the near wake (Jeon et al., 2016), rotor performance on a moving platfrom (Dong et al., 2019), wake dynamics for specific motions (Lee and Lee, 2019), and control optimisation for a coupled aero- and hydrodynamic model (van den Broek et al., 2022c).





Most FVW models focus on wake dynamics close to the rotor and, to our best knowledge, little validation has been done for the mid to far wake. We define the mid wake from $1D$ to $4D$ and the far wake beyond $4D$ downstream from the rotor, where $D$ is the rotor diameter. Therefore, this study aims to validate the applicability of FVW models as surrogate models for dynamic wind farm flow control optimisation, especially focused on wake steering through yaw misalignment. For that purpose, we consider three different turbine model formulations: the three-dimensional actuator-disc model, an extension of the actuator disc with a root vortex to model wake rotation, and an extension to an actuator-line model (van Kuik, 2018). This simplified actuator-line model is also known as the Joukowsky rotor.

The validity of these models for wake predictions under yaw misalignment is evaluated with three sets of experimental data from wind tunnel measurements. First, a set of data that consists of lidar measurements of wind turbine wakes under steady yaw misalignment (Hulsman et al., 2022b). Second, a set of time-resolved particle-image velocimetry measurements of the wake following a step change in yaw angle (van der Hoek et al., 2023a). Third, turbine output measurements from an experiment for yaw-based wake steering with wind direction variation (Hulsman et al., 2023).

The contribution of this paper is twofold: (i) an analysis of model parameter choice and suitable levels of simplification for representing the wind turbine wake, and (ii) a validation of free-vortex wake models for mid- to far-wake power predictions with wind tunnel data, in light of control optimisation for yaw control.

The remainder of this paper is structured as follows. Section 2 presents the free-vortex method and develops the three different models of the wind turbine wake, followed by a study of parameter sensitivity and convergence in Sect. 3. The data sets from the wind tunnel experiments and methods for validation are presented in Sect. 4. The results are then discussed in Sect. 5 and, finally, the conclusions are shown in Sect. 6.

## 2 Free-vortex wake models

First, we construct the models of the wind turbine wake that are studied in this paper. The free-vortex methods and straight-line vortex filament definition are introduced in Sect. 2.1. Using these filaments as building blocks, the three wind turbine representations for wake modelling for control optimisation are then described in Sect. 2.2.

### 2.1 Free-vortex wake

The basis of the vortex methods is the vorticity formulation of the Navier-Stokes equations. The FVW method is based on Lagrangian particles that advect downstream. These particles induce a velocity based on their associated circulation strength. The resultant flow velocity may be calculated at any position based on the free-stream velocity and the sum of induced velocities. The vorticity formulation requires the assumption of inviscid and incompressible flow, although diffusion may be approximated. For a further description of the fundamentals, the reader is referred to aerodynamic literature, such as Leishman (2000) or Katz and Plotkin (2001).





### 2.1.1 Vortex filaments

The three-dimensional model formulations in this study are based on straight-line vortex filaments. The induced velocity $\boldsymbol{u}_\mathrm{i} \in \mathbb{R}^3$ at a point $\boldsymbol{x}_0 \in \mathbb{R}^3$ is calculated with the Biot-Savart law from a single vortex filament starting at $\boldsymbol{x}_1 \in \mathbb{R}^3$ and ending at $\boldsymbol{x}_2 \in \mathbb{R}^3$, with vortex strength $\Gamma$,

$$\boldsymbol{u}_\mathrm{i}(\boldsymbol{x}_0, \boldsymbol{x}_1, \boldsymbol{x}_2) = \frac{\Gamma}{4\pi} \frac{\boldsymbol{r}_1 \times \boldsymbol{r}_2}{||\boldsymbol{r}_1 \times \boldsymbol{r}_2||^2} \boldsymbol{r}_0 \cdot \left( \frac{\boldsymbol{r}_1}{||\boldsymbol{r}_1||} - \frac{\boldsymbol{r}_2}{||\boldsymbol{r}_2||} \right), \tag{1}$$

where the relative positions $\boldsymbol{r} \in \mathbb{R}^3$ are defined as

$$\boldsymbol{r}_0 = \boldsymbol{x}_2 - \boldsymbol{x}_1, \tag{2}$$
$$\boldsymbol{r}_1 = \boldsymbol{x}_1 - \boldsymbol{x}_0, \tag{3}$$
$$\boldsymbol{r}_2 = \boldsymbol{x}_2 - \boldsymbol{x}_0. \tag{4}$$

A Gaussian core with core size $\sigma$ is included to regularise singular behaviour of the induced velocity close to the vortex filament.

$$\boldsymbol{u}_{\mathrm{i},\sigma}(\boldsymbol{x}_0, \boldsymbol{x}_1, \boldsymbol{x}_2) = \boldsymbol{u}_\mathrm{i} \left( 1 - \exp\left( -\frac{||\boldsymbol{r}_1 \times \boldsymbol{r}_2||^2}{\sigma^2 ||\boldsymbol{r}_0||^2} \right) \right). \tag{5}$$

### 2.1.2 Convection of vortex filaments

Vortex filaments are convected over time according to the combination of the free-stream velocity $\boldsymbol{u}_\infty \in \mathbb{R}^3$ and the total velocity induced by all filaments $\boldsymbol{u}_\mathrm{ind} \in \mathbb{R}^3$ at the vortex position $\boldsymbol{x} \in \mathbb{R}^3$,

$$\dot{\boldsymbol{x}} = \boldsymbol{u}_\mathrm{ind}(\boldsymbol{x}) + \boldsymbol{u}_\infty(\boldsymbol{x}), \tag{6}$$

where $\dot{\boldsymbol{x}} \in \mathbb{R}^3$ is the time derivative of the vortex position. At fixed intervals, a new set of vortex filaments is released from the rotor according to the wind turbine model definition. The oldest set of vortex filaments is then discarded from the simulation, such that a fixed number of sets of filaments is maintained.

### 2.1.3 Modelling viscous diffusion

Turbulence is not explicitly accounted for when using the FVW to construct models of wind turbine wakes. However, growth of the vortex core may be used to approximate the effects of turbulent and viscous diffusion as

$$\sigma_{k+1} = \sqrt{4\alpha\delta\nu\Delta t + \sigma_k^2}, \tag{7}$$

which is Squire's modification of the diffusive growth of Lamb-Oseen vortex core (Squire, 1965), with the discrete time step $k$, the constant $\alpha = 1.25643$, effective turbulent viscosity coefficient $\delta$ to tune core growth, kinematic viscosity $\nu = 1.5 \times 10^{-5}\,\mathrm{m^2\,s^{-1}}$, and time step $\Delta t$.



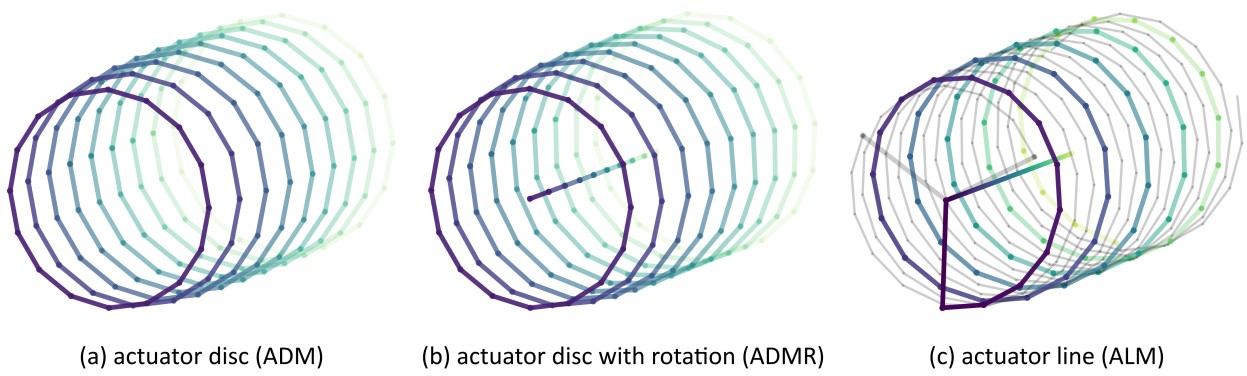

(a) actuator disc (ADM)    (b) actuator disc with rotation (ADMR)    (c) actuator line (ALM)

**Figure 1.** The vortex filament structures for the three different free-vortex wind turbine representations under consideration: (a) the actuator disc (ADM) builds up a wake with discretised vortex rings, (b) the actuator disc with rotation (ADMR) adds a root vortex along the wake centre to model swirl due to turbine rotation, and (c) the actuator line (ALM) models individual blade circulation, tip vortices, and a combined root vortex. In (c), the vortex structure from a single blade is highlighted for clarity.

## 2.2 Wind turbine models

The wind turbine models used for this study are the three-dimensional actuator disc as used by van den Broek et al. (2022a), an
extension with rotation, and an actuator-line model. These three concepts are illustrated in Fig. 1. Note that a two-dimensional actuator disc could be considered as a further simplification of the wind turbine wake under axisymmetric conditions. It is however not considered in the current study because it has already been shown to be ineffective in modelling the wind turbine wake under yaw misalignment due to a lack of axisymmetry (van den Broek et al., 2022a).

### 2.2.1 Coefficients for wind turbine modelling

The turbine thrust $T$ is calculated according to the magnitude of the free-stream inflow velocity $u_\infty$,

$$T = c_\mathrm{t} \cdot \frac{1}{2} \rho A_\mathrm{r} u_\infty^2 \cos^{\beta_\mathrm{t}}(\gamma) \,, \tag{8}$$

with thrust coefficient $c_\mathrm{t}$, air density $\rho$, and rotor swept area $A_\mathrm{r}$. The thrust is assumed to vary over yaw misalignment $\gamma$ with a cosine-exponent $\beta_\mathrm{t}$ which may be adjusted to represent experimental data. Similarly, aerodynamic power $p_\mathrm{a}$ is calculated with the power coefficient $c_\mathrm{p}$ as

$$p_\mathrm{a} = c_\mathrm{p} \cdot \frac{1}{2} \rho A_\mathrm{r} u_\infty^3 \cos^{\beta_\mathrm{p}}(\gamma) \,, \tag{9}$$





where the exponent $\beta_\mathrm{p}$ can be used to tune power variation over yaw misalignment, such as seen in Hulsman et al. (2022a). The thrust and power coefficient are assumed to be a function of the induction factor $a$ based on momentum theory,

$$
c_\mathrm{t}(a) = \begin{cases} 4a(1-a) & \text{if } a \leq a_\mathrm{t}\,, \\ c_\mathrm{t1} - 4(\sqrt{c_\mathrm{t1}}-1)(1-a) & \text{if } a > a_\mathrm{t}\,, \end{cases} \tag{10}
$$

$$
c_\mathrm{p}(a) = 4a(1-a)^2\,, \tag{11}
$$

with the parameter $c_\mathrm{t1} = 2.3$ and the induction at the transition point $a_\mathrm{t} = 1 - \frac{1}{2}\sqrt{c_\mathrm{t1}}$ . The thrust coefficient calculation is based on momentum theory with a transition to a linear approximation for high induction values, which is an empirical correction based on the Glauert correction (Burton et al., 2001).

### 2.2.2 Actuator disc (ADM)

An actuator-disc representation of a wind turbine is implemented with the free-vortex method and illustrated in Fig. 1(a). The
actuator disc is assumed to be uniformly loaded so it only sheds vorticity along its edge (Katz and Plotkin, 2001).

At fixed time intervals, a vortex ring discretised in $n_\mathrm{e}$ vortex filaments is generated at the edge of the rotor. The vorticity $\Gamma$ generated along the edge of an actuator disc is directly related to the pressure differential generated by the disc (van Kuik, 2018). This pressure jump is due to the thrust force,

$$
\Gamma = \Delta t \frac{\partial \Gamma}{\partial t} = \Delta t \frac{1}{\rho} \frac{T}{A_\mathrm{r}}\,. \tag{12}
$$

### 2.2.3 Actuator disc with rotation (ADMR)

An extension of the ADM is the actuator disc with rotation (ADMR). A root vortex is released along the centre-line of the wake as shown in Fig. 1(b). Note that the associated distributed vorticity over the disc and the wake boundary is neglected. This root vortex models the swirl in the wake induced by the rotation of the wind turbine rotor. The inclusion of rotation may contribute to modelling asymmetry in wake steering.

Assuming again that the rotor is uniformly loaded, the thrust force is equally distributed over each of the blades,

$$
L = \frac{T}{n_\mathrm{b}}\,, \tag{13}
$$

where $L$ is the individual blade loading and $n_\mathrm{b}$ is the number of blades. The bound vorticity at the blade $\Gamma_\mathrm{b}$ is then calculated according to the Kutta-Joukowsky theorem,

$$
\frac{\partial L}{\partial r} = -\rho u_\mathrm{rel}(r)\Gamma_\mathrm{b} = -\rho u_\infty \omega r \Gamma_\mathrm{b}\,, \tag{14}
$$

with the relative velocity $u_\mathrm{rel}$ along the blade $r$ with rotational velocity $\omega$. Integration over the blade length then yields

$$
L = \rho \lambda u_\infty \frac{1}{2} R \Gamma_\mathrm{b}\,, \tag{15}
$$





**Table 1.** Numerical parameters for the FVW models as used for this validation study.

|  |  | ADM | ADMR | ALM |
|---|---|---|---|---|
| time step | $\Delta t \cdot u_\infty / D$ | 0.3 | 0.3 | 0.033 |
| number of rings | $n_\mathrm{r}$ | 40 | 40 | 360 |
| elements per ring | $n_\mathrm{e}$ | 16 | 17 | 7 |
| initial core size | $\sigma / D$ | 0.16 | 0.16 | 0.16 |
| turbulent growth | $\delta$ | 100 | 100 | 100 |

where $\lambda$ is the tip-speed ratio and $R$ the rotor radius. The bound vortex strength of a single blade is then

$$\Gamma_\mathrm{b} = \frac{c_\mathrm{t} u_\infty \pi R}{n_\mathrm{b} \lambda}. \tag{16}$$

Combining circulation of the bound vortices of each blade yields the root vortex strength $\Gamma_\mathrm{r}$,

$$\Gamma_\mathrm{r} = n_\mathrm{b} \Gamma_\mathrm{b}. \tag{17}$$

#### 2.2.4 Actuator line (ALM)

The Joukowsky rotor model is an actuator-line model (ALM) that assumes uniform blade loading, forming a rotating horse-shoe vortex system for each blade (van Kuik, 2018). The vortex filament structure is shown in Fig. 1(c), where the vortex system from a single blade is highlighted. Each blade is modelled with a bound vorticity $\Gamma_\mathrm{b}$ as in Eq. (16). The tip vortices coming off from each of the blades have the same vorticity, $\Gamma_\mathrm{t} = \Gamma_\mathrm{b}$. The root vortex is the combination of the bound vorticity of each of the blades, $\Gamma_\mathrm{r} = n_\mathrm{b} \Gamma_\mathrm{b}$, which is equivalent to the one previously introduced in the ADMR.

### 3 Parameter study and convergence

An important aspect of the wind turbine models from Sect. 2.2 is the sensitivity to parameter changes. This section explores the convergence behaviour of the ADM with the aim of finding a set of suitable parameters to be used in the comparison with wind tunnel data in Sect. 4. For brevity, the convergence behaviour of the ADMR and ALM is performed but not included, as the results are similar to the ADM. The reference parameter values for all three models are listed in Table 1. The study is split into four parts: first, the effect of the streamwise spatial discretisation in Sect. 3.1, second, time discretisation in Sect. 3.2, third, the sensitivity to the azimuthal spatial discretisation of the vortex rings in Sect. 3.3, and, fourth, the effect of the core size in Sect. 3.4.

### 3.1 Streamwise spatial discretisation

The streamwise spatial discretisation of the ADM is studied by constructing a cylindrical vortex tube of length $12\,D$ from discretised vortex rings, approximating the ADM wake. The spacing between vortex rings is varied to study the effect on the

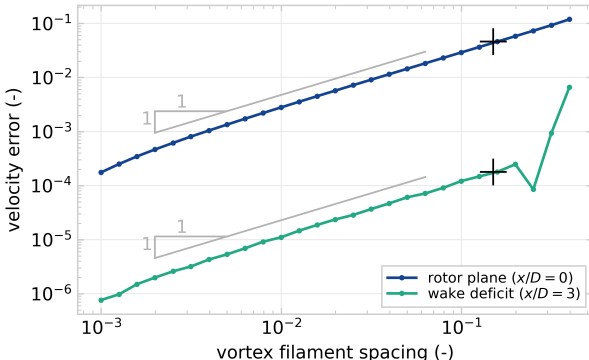

**Figure 2.** Relative error of velocity deficit at the rotor plane $x/D = 0$ and at a downstream distance $x/D = 3$. Spacing between vortex rings is varied and circulation strength of the filaments is adjusted accordingly. Reference solution for spacing $\Delta x/D = 5 \times 10^{-4}$. The crosses '+' mark the approximate streamwise spacing produced for $\Delta t \cdot u_\infty / D = 0.3$ as used throughout this paper.

wake deficit without the effects of temporal evolution. The number of rings is adjusted accordingly to maintain a constant wake length and the circulation of the vortex filaments is adjusted to maintain the same distribution of total circulation. The velocity

error $\varepsilon_\mathrm{u}$ is defined as

$$\varepsilon_\mathrm{u}(x) = |u(x) - u_\mathrm{ref}(x)|, \tag{18}$$

where $u$ is the induced velocity on the wake centre line and $u_\mathrm{ref}$ is a reference value, which is generated for a spacing of $\Delta x/D = 5 \times 10^{-4}$.

The convergence behaviour of the velocity deficit with increasing number of rings is first order as is illustrated in Fig. 2. The

variation of error over filament spacing within the wake, at $x/D = 3$, is small compared to the variation of error at the entry of the tube, which corresponds to the rotor plane, $x/D = 0$. The sharp increase in error for $x/D = 3$ for the coarsest spacing is caused by an insufficiently large core size, which produces an oscillating velocity profile in the wake. This is not an issue as the core size $\sigma/D = 0.16$ used in the rest of this paper produces a smooth velocity deficit profile for the chosen numerical settings, i.e., there are no significant oscillations in velocity magnitude along the wake centre streamline. The inflow at the rotor disc

varies a lot for different filament spacing values, which limits the consistency of using local velocity measurements at the rotor plane.

It is important to note that streamwise spatial discretisation is directly connected to the time discretisation and the computational complexity. The largest possible time step is such that a vortex ring is released at every time step. High spatial resolution is thus only possible for small time steps. Additionally, the large number of elements required to generate a wake of sufficient

length with high streamwise resolution leads to large increases in the computational cost of the induced velocity calculation; the cost of the induced velocity evaluation increases quadratically with the number of vortex filaments. Small time steps and expensive induced velocity calculation both contribute to a significant increase in computational cost for a given prediction

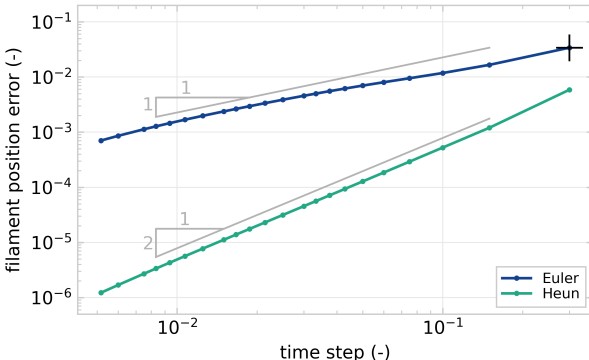

**Figure 3.** Relative error of the position of the vortex filaments for varying time discretisation, comparing the first order explicit Euler method and the second order explicit Heun's method. Reference solution for time step $\Delta t \cdot u_\infty / D = 1.5 \times 10^{-3}$. The cross '+' marks $\Delta t \cdot u_\infty / D = 0.3$ as used throughout this paper.

horizon. Therefore, a relatively large time step and coarse spatial resolution are chosen for the purpose of efficient optimisation of wind turbine controls.

195 The ADMR introduces a single extra vortex filament per time step compared to the ADM, which makes it about $1.1\times$ more expensive with the current numerical settings. The ALM releases requires four times as many filaments as the ADM for a wake of the same length, which makes a single time step sixteen times more expensive. Accounting for the smaller time steps, simulating a given time with the ALM is about $140\times$ more expensive than the ADM.

### 3.2 Time discretisation

200 The time discretisation of the FVW is studied by examining convergence for a first and a second order integration scheme. In order to perform this convergence experiment, it is necessary to decouple streamwise spatial discretisation and time discretisation. We reformulate the problem such that a number of sub-steps may be taken between releasing vortex rings.

 The largest time step considered is $\Delta t \cdot u_\infty / D = 3 \times 10^{-1}$, where one set of vortex filaments is released at every step. From there, the time step is reduced to $5 \times 10^{-3}$ and a reference solution is generated with a step size of $\Delta t \cdot u_\infty / D = 1.5 \times 10^{-3}$.

205 The convergence is then quantified using the mean position error $\varepsilon_{\mathrm{x}}$ of all vortex filaments with respect to the reference solution,

$$\varepsilon_{\mathrm{x}} = \mathrm{mean}\left(\|\boldsymbol{x}_i - \boldsymbol{x}_{\mathrm{ref},i}\|_2\right) \quad \text{for } i = \{1, 2, \ldots, n_{\mathrm{p}}\}, \tag{19}$$

where $\boldsymbol{x}_i$ are the $n_{\mathrm{p}}$ coordinates defining the positions of the vortex filaments and $\boldsymbol{x}_{\mathrm{ref},i}$ is the reference solution.

 Figure 3 shows the convergence of time integration for decreasing step size with the first order explicit Euler method as used in this paper and with the second order explicit Heun's method for comparison. For the numerical parameters presented

210 here, the methods converge as expected. The chosen time step $\Delta t \cdot u_\infty / D = 0.3$ is rather large because of the emphasis on computational efficiency for control optimisation. This is also the reason for choosing explicit Euler, as it requires only a



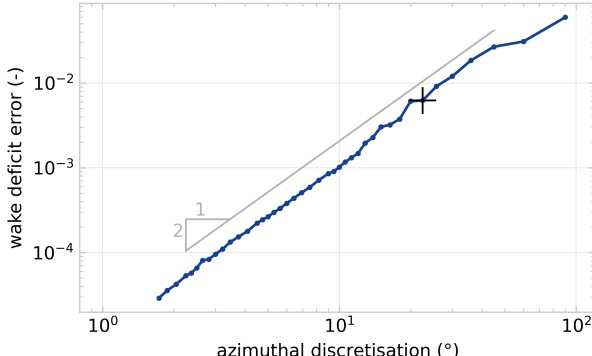

**Figure 4.** Relative error of the cross-stream rotor-averaged velocity profile at $5\,\mathrm{D}$ downstream for the ADM under a yaw misalignment of $\gamma = 30°$. Azimuthal discretisation is $360°/n_\mathrm{e}$ and the reference is at $0.7°$ for $n_\mathrm{e} = 512$ elements. The cross '+' marks $22.5°$ for $n_\mathrm{e} = 16$ as used throughout this paper.

single function evaluation per time step. If a higher degree of convergence is required from the time integration, a change of integration method is more efficient than reduction of time step.

### 3.3 Azimuthal spatial discretisation

The convergence of azimuthal discretisation is tested by varying the number of elements in the vortex rings. A simulation with a yaw misalignment of $30°$ under steady inflow is performed for the different discretisation steps. When the initial transient of the developing wake is passed, a cross-stream profile of rotor-averaged velocity is recorded. The error norm of this deficit profile $\varepsilon_\mathrm{d}$ is

$$\varepsilon_\mathrm{d} = \mathrm{mean}(|u_\mathrm{r}(y) - u_\mathrm{r,ref}(y)|) \quad \text{for} \ -2 < y/D < 2, \tag{20}$$

where $u_\mathrm{r}(y)$ is the rotor-averaged velocity at cross-stream position $y$ and the reference solution is generated for $n_\mathrm{e} = 512$ elements in the ring discretisation.

Figure 4 shows that the velocity deficit profile converges for increasing number of vortex filaments in the vortex ring discretisation. The azimuthal discretisation in the current study is for $n_\mathrm{e} = 16$.

The time discretisation of the ALM is chosen such that it achieves the same azimuthal resolution, which is for $\Delta t \cdot u_\infty/D = $ 225 $0.033$. A time step that is nine times smaller implies nine times as many vortex rings – the set of vortex filaments released at one time step – are necessary to model a wake of the same length as the ADM. The ALM thus combines a smaller time step and a more expensive velocity calculation due to the larger number of vortex filaments, which makes it less attractive for control optimisation for long wakes.



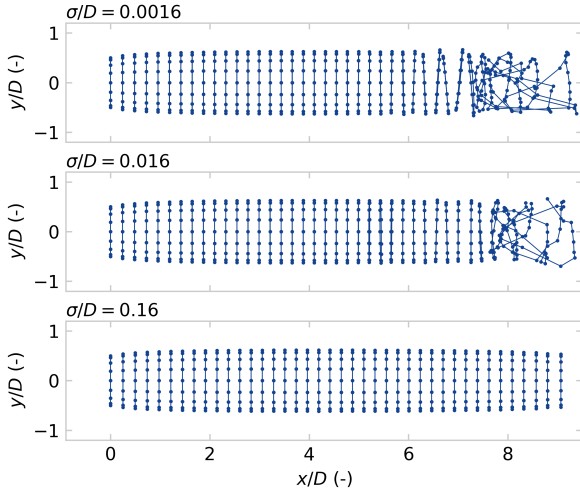

**Figure 5.** Illustration of vortex particle/filament trajectories for varying initial core size. Larger core sizes produce more stable – less unstable – trajectories. The results in this paper are produced for an initial core size $\sigma/D = 0.16$.

### 3.4 Vortex core size

The choice of vortex core size $\sigma$ plays a major role in the stability of the FVW models as illustrated for the ADM in Fig. 5 for different sizes of the Gaussian vortex core. For small constant core sizes, the wake structure naturally transitions into instability, leading to chaotic development of the wake downstream. For larger core sizes, this disturbance growth is smoothed out and the wake structure appears more stable. The size of the vortex core needs to be tuned to the streamwise spatial resolution of the wake. It should at least be large enough to guarantee a smooth velocity profile between vortex filaments, avoiding oscillations

in the wake deficit. On the other hand, it should be small enough not to lose information.

Variation of the vortex core size has very little influence on the initial wake depth. However, the wake recovery can be tuned with vortex growth, implemented with Eq. (7). Figure 6 shows how increasing the turbulent core growth parameter $\delta$ impacts the recovery of the wake and allows tuning the representation of turbulent mixing. The validation simulations run in this paper are for $\delta = 100$, which is chosen to model some wake recovery.

### 4 Validation with wind tunnel data


Following the study of numerical model parameters, it is essential to validate the wake flow and power predictions of the FVW model for yaw control optimisation. This section first presents the available data from the three wind tunnel experiments in Sect. 4.1. The performance measures used to quantify performance are then introduced in Sect. 4.2, followed by details on the replication of the experiments with the FVW models in Sect. 4.3.





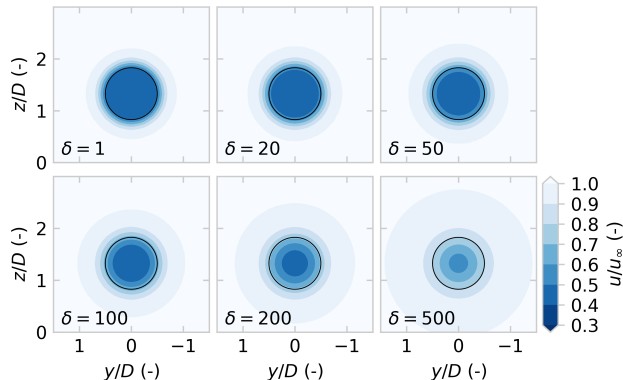

**Figure 6.** Introducing vortex core growth using the Lamb-Oseen model allows tuning of the diffusion to approximately match turbulent mixing in the wake. Slices at $x/D = 5$ downstream for turbulent growth parameter $\delta$ from 1 to 500, with an initial core size $\sigma/D = 0.16$.

## 4.1 Wind tunnel experiments

Three sets of experimental data are used in this paper for the model validation study. The first is a set of steady state flow measurements for the wind turbine wake under yaw misalignment (Hulsman et al., 2022b). The second is a dynamic experiment with high temporal resolution of a step change in yaw angle (van der Hoek et al., 2023a). The third is a longer set of turbine output measurements for wake redirection with wind direction variation. (Hulsman et al., 2023)

The wind direction $\theta$ is defined clockwise positive, with $0°$ along the positive $x$-axis pointing downstream. The yaw angle $\psi$ is clockwise positive with a $180°$ shift such that the rotor is fully aligned with the wind direction if $\psi = \theta$. The yaw misalignment $\gamma = \theta - \psi$, such that a counter-clockwise misalignment is positive.

All experiments used the MoWiTO-0.6 turbine with a rotor diameter $D = 0.58\,\mathrm{m}$ (Schottler et al., 2016). The model turbine has pitch control and the generator can be used for torque control to regulate rotor speed.

### 4.1.1 Steady yaw misalignment – WTA

The first set of experimental data (labelled WTA) was recorded in the wind tunnel at ForWind, University of Oldenburg and has been published in (Hulsman et al., 2022b). It provides measurements of wind turbine wakes under yaw misalignment with steady flow conditions.

The data contains measurements of wakes for yaw misalignment angles $\gamma = \{-30°, 0°, 30°\}$. The turbines were mounted at a hub height of $0.77\,\mathrm{m}$. Operation was at a tip-speed ratio of $\lambda = 5.7$ for aligned flow and $\lambda = 5.3$ when misaligned.

The wind tunnel has a $3\,\mathrm{m} \times 3\,\mathrm{m}$ test section with an active grid, used passively, to control inflow turbulence and boundary layer profiles.

The cross-sectional averaged flow measurements were obtained with a WindScanner lidar performing a Lissajous scan within a $3D \times 3D$ area, for uniform and sheared inflow with a turbulence intensity of around $1\,\%$. Wind speed was $7.5\,\mathrm{m\,s^{-1}}$ at hub



height. Only the uniform inflow data is analysed in the current study, primarily the vertical measurement planes at downstream distances from $x/D = 1$ to 7.

### 4.1.2 Step change in yaw – WTB

The second set of experimental data (labelled WTB) was recorded in the Open Jet Facility (OJF) at the TU Delft, with the same set-up as used in (van der Hoek et al., 2023a). A step change in yaw was measured at high temporal resolution using tomographic particle-image velocimetry (PIV).

A PIV set-up with four cameras was used to measure the flow velocity at downstream distances $x/D = \{1, 2, 3, 4\}$. The measurements were recorded at $500\,\text{Hz}$ for a $5\,\text{s}$ duration. The step change was initiated $1\,\text{s}$ after a trigger signal. This trigger also initiated the PIV measurements and was used to synchronise the data at the four downstream distances.

The turbine yawed from a $\gamma = 0°$ to a $\gamma = 20°$ misalignment with the inflow, at a yaw rate of approximately $\dot{\psi} = 16°\text{s}^{-1}$. The turbine operated at a tip-speed ratio of $\lambda = 5.5$ for an inflow velocity $u_\infty = 4.7\,\text{m\,s}^{-1}$.

### 4.1.3 Wind direction variation – WTC

The third set of experimental data (labelled WTC) was again recorded in the wind tunnel at ForWind, University of Oldenburg (Hulsman et al., 2023). Instead of steady yaw misalignment, it provides turbine output measurements of a dynamic yaw experiment. Two turbines were used to test open-loop yaw control strategies under time-varying wind direction.

The upstream turbine was stationary and yaw-controlled to achieve wake steering. The second turbine was placed $2.66\,D$ downstream with an increased hub height of $0.16\,D$. The downstream turbine is translated on an $xy$-traverse system to model wind direction variation. The active grid was used passively to generate inflow with a turbulence intensity $\text{TI} < 1\,\%$ with a shear profile with shear exponent $\alpha = 0.28$, where the wind speed was $7.3\,\text{m\,s}^{-1}$ at hub height for the upstream turbine.

The yaw setpoints for the upstream turbine were stored in a look-up table and applied differently for each control experiment. The experiments are labelled BW30, BW60, BW120, BW300, BW600 based on the length of wind direction averaging window used in the controller, with shorter windows leading to more frequent yaw variations. Figure 7 illustrates the variation of yaw angle and relative wind direction for the BW30 experiment. Each controller experiment yielded $10\,\text{min}$ of turbine data, such as generator power, torque, and rotor speed, recorded with a $5\,\text{kHz}$ sampling frequency. The raw data has been filtered with a low-pass filter with a cut-off frequency at $20\,\text{Hz}$ for noise reduction before use in the current study.

## 4.2 Performance measures

The performance measures in this study reflect the purpose of this model. It is oriented towards control for power maximisation and therefore the predictive qualities for wake deflection and downstream aerodynamic power availability are important aspects to measure.

Wake deflection is determined according to the wake centre position, which is defined as the cross-stream position where aerodynamic power available for a virtual rotor at hub height is minimal. The potential power follows quite directly from the





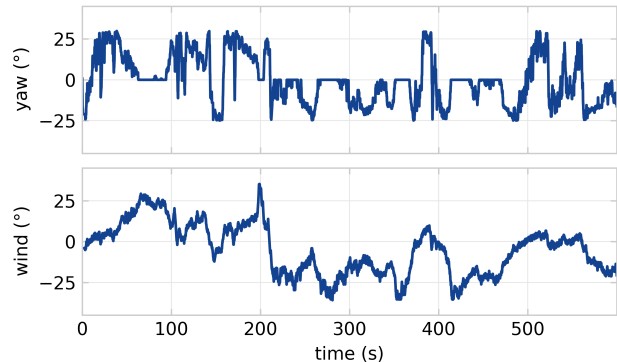

**Figure 7.** Time series data from WTC showing yaw misalignment on the upstream turbine and the wind direction variation for the BW30 experiment. The changes in wind direction are achieved by translation of the downstream turbine on an $xy$-traverse system.

measured or simulated flow field and is directly related to downstream turbine performance as available aerodynamic power $p^*$ is calculated from the rotor-averaged velocity $\boldsymbol{u}_{\mathrm{r}}$,

$$p^* = c_{\mathrm{p}}^* \frac{1}{2} \rho A_{\mathrm{r}} \left( \boldsymbol{n} \cdot \boldsymbol{u}_{\mathrm{r}} \right)^3 \tag{21}$$

where $c_{\mathrm{p}}^*$ is the theoretical maximum power coefficient and $\boldsymbol{n}$ is the unit vector orthogonal to the rotor plane. This is similar to

300   the potential power method introduced in (Schottler et al., 2018).

For statistical analysis, the fit of the power predictions are evaluated with the variance accounted for (VAF),

$$\mathrm{VAF} = \left( 1 - \frac{\mathrm{var}(p - \hat{p})}{\mathrm{var}(p)} \right) \cdot 100\,\%, \tag{22}$$

and the normalised mean absolute error (NMAE)

$$\mathrm{NMAE} = \frac{\mathrm{mean}(|p - \hat{p}|)}{\mathrm{max}(p)} \cdot 100\,\%, \tag{23}$$

305   where $\hat{p}$ is the predicted power from the FVW model and $p$ is the measured power from the wind tunnel. Note that VAF values closer to $100\,\%$ indicate better performance, whereas NMAE values closer to $0\,\%$ represent a close fit.

A total comparison of power at different yaw angles is performed by binning the results according to wind direction and yaw angle bins and calculating mean and standard deviation of the power in each bin. The analysis is the same for both model and experimental data, thus allowing equivalent comparison.

310   **4.3   Experiment replication**

A visual, qualitative comparison of the available flow measurements provides a general overview of the strengths and weaknesses of each of the models. These are provided for steady-state measurements from WTA and WTB. The cross-stream planes of the flow are considered to be more important than hub-height planes because of the three-dimensional nature of wind turbine wakes under yaw misalignment.



A quantitative analysis of the steady-state wake deflection from WTA is performed by analysis of the flow cross-sections for the cross-stream position where potential power is minimal. This is considered to be the wake centre and a measure for predictive power for wake deflection under steady-state conditions.

The WTB experiment is replicated with all three FVW models to analyse the temporal dynamics of the model at high time resolution. The time series of potential power production provide insight into the propagation of yaw effects downstream through the wake.

Finally, the dynamic experiment in WTC is fully replicated with the ADM free-vortex wake model. The upstream turbine is set to the specified yaw angle as recorded in the experiment data and operated under a constant thrust coefficient. The downstream turbine performance is evaluated from the rotor-averaged velocity over a rotor disc at the downstream turbine position. This position varies over time as the turbine is translated to track the specified wind direction from the experiment. For both upstream and downstream turbines, the rotor-averaged velocity is recorded. The downstream velocity is increased by $2.5\%$ to account for the increased velocity due to the higher hub height in the shear profile of the inflow.

The wake model provides an estimate of the velocity but the experimental data records generator power. A simple turbine model is specified to account for inertial dynamics,

$$\omega_{k+1} = \omega_k + \frac{\Delta t}{J} \left( p_\mathrm{a} - \tau \omega_k \right), \tag{24}$$

where $\omega_k$ is the angular velocity of the rotor at time step $k$, $\Delta t$ is the time step size, $J$ is the rotor inertia, $p_\mathrm{a}$ is the available aerodynamic power and $\tau$ is the generator torque. Generator torque $\tau$ is calculated from the angular velocity of the rotor $\omega$ as

$$\tau = k_1 \omega + k_2 \omega^2, \tag{25}$$

with the gains $k_1$ and $k_2$. The form of this control law is based on the turbine controller used in the experiment, which was developed by Petrović et al. (2018). Generator torque and angular velocity are multiplied with an efficiency $\eta$ to obtain a generator power estimate,

$$p_\mathrm{g} = \eta \tau \omega. \tag{26}$$

The efficiency term is there to capture all inefficiencies in converting aerodynamic power to electrical power, such as a suboptimal power coefficient and drive train losses. The parameters for the turbine model and controller polynomial are found through a least-squares estimate. The controller gains are first estimated using measured rotor speed and generator torque. The rotor inertia and power conversion efficiency are then estimated using the torque controller, modelled wind speed, and measured power at the upstream turbine. The yaw dependency of thrust and power is tuned with respective cosine-exponents $\beta_\mathrm{t} = 1$ and $\beta_\mathrm{p} = 2$. The yaw-aligned thrust coefficient for all experiments is set to $c_\mathrm{t} = 0.9$ for $a = 0.33$. The final power estimate is filtered with the same low-pass filter that is applied to the experimental data.



## 5 Results and discussion

Now, this section presents the core results of this paper, the comparison of the model predictions with wind tunnel data for validation. Yaw misalignment under steady conditions is discussed in Sect. 5.1, followed by the time-resolved step change in yaw in Sect. 5.2. Finally, analysis of the wake steering experiment with wind direction variation is provided in Sect. 5.3.

### 5.1 Steady yaw misalignment – WTA

A visual comparison of cross-stream wake velocity profiles is provided in Fig. 8, illustrating wake development under yaw
misalignment angles $\gamma = \{-30°, 0°, 30°\}$ for experimental data from WTA and simulation results with the ADM, ADMR, and ALM.

For yaw-aligned flow ($\gamma = 0°$), the wake is almost axisymmetric, both in the wind tunnel as well as in the FVW models. The wakes in the FVW are stable with a consistent wake deficit. There is some underestimation of wake depth in the near wake with the current numerical parameter choice as listed in Table 1. Some recovery is modelled through the growth of the Gaussian
core as descibed in Sect. 3.4 to represent the turbulent mixing that is visible in the wind tunnel measurements.

Both positive and negative yaw misalignment angles result in the generation of a counter-rotating vortex pair and subsequently a curled wake shape, which becomes apparent from $3\,D$ downstream. The ADM produces wakes that are symmetric between positive and negative yaw misalignment, as expected. The inclusion of the root vortex in the ADMR models some of the asymmetry in wake shape that is also present in the experimental data. The ALM produces a similar asymmetric deforma-
tion of the wake.

A large deformation of the wake is visible from $5\,D$ onwards for the wake under yaw misalignment. In the FVW models, this leads to stretching of the vortex filaments and unstable wake structures. Especially, the wake of the ALM breaks down beyond $5\,D$ downstream because of the large number of vortex filaments in close proximity. The ADMR still demonstrates similar stability in wake structure as the ADM at $7\,D$ downstream, although resemblance of the wind tunnel data is reduced.

Finally, some of the details of the experimental data are not represented in the FVW models. The effect of the wake from the turbine tower on power predictions is assumed to be minor and, therefore, it is not considered in the FVW models, although it is present in the wind tunnel measurements. Additionally, the effect of the ground is neglected, whereas the experimental data shows a thin boundary layer near the bottom of the wind tunnel. Ground effects in FVW models may be modelled using a mirrored vortex structure (Leishman, 2000). An initial experiment showed that some asymmetry in wake deflection may be
achieved this way. However, for the sake of limiting computational complexity, this option is not presented in this study.

The displacement of the wake centre is evaluated according to the cross-stream position where available aerodynamic power is minimal. Figure 9 compares the wake deflection for the data from WTA and the three FVW models. This corresponds to the wakes shown in Fig. 8.

The ADM appears to have the best fit to the experimental data over the measured downstream distances. The ADMR
shows a similar deflection profile up to $5\,D$ downstream. The ALM only shows good agreement with the experimental data up to $x/D = 5$, which matches the visual analysis of the wake structure. Especially towards $7\,D$ downstream, the wake centre



**Figure 8.** A comparison of normalised streamwise velocity for wind turbines under yaw misalignment. The experimental data from WTA are compared to model results with the ADM, ADMR, and ALM with slices at $x/D = \{1, 3, 5, 7\}$ for yaw misalignment angles $\gamma = \{-30°, 0°, 30°\}$. All three models represent the curled shape of the wake under yaw misalignment. The inclusion of rotation (ADMR, ALM) improves the qualitative representation of the asymmetry and vertical displacement that is observed in the wind tunnel. The deformation of the vortex filament structure with the ALM beyond $x/D = 5$ becomes too large to provide useful predictions.



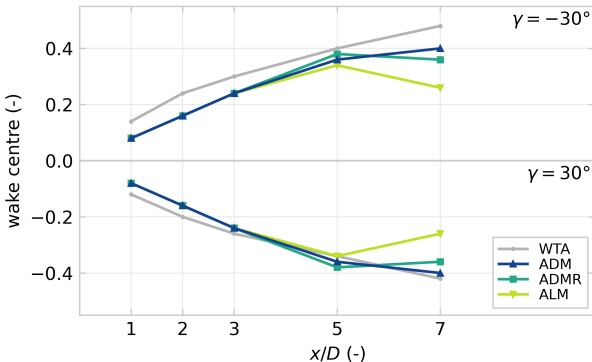

**Figure 9.** Wake centre deflection over downstream distance calculated from the flow slices illustrated in Fig. 8, where the wake centre position is defined as the cross-stream position with minimal available aerodynamic power. The ADM provides the best estimate of wake centre deflection, whereas the rotation in the ADMR and ALM models leads to divergence of the deflection estimate by $x/D = 7$.

predictions diverge. This is related to the large deformation of the wake under these high yaw angles and the numerical stability of the FVW wake structures. The lack of rotation in the ADM allows for a more stable wake structure and a better prediction of the wake deflection further downstream.

Sufficiently accurate wake predictions and low computational cost are important aspects of a control-oriented wake model. In this respect, we observe that the extensions of the FVW models beyond the ADM do not appear to contribute to better predictions of wake deflection under yaw misalignment and that the ADM is computationally the cheapest to evaluate. Therefore, the replication of dynamic experiments will focus only on the ADM.

### 5.2 Step change in yaw – WTB

Experiment WTB recorded wake development at downstream distances $x/D = \{1, 2, 3, 4\}$ for a turbine yawing from $\gamma = 0°$ to $\gamma = 20°$. The cross-stream wake velocity profiles for $x/D = \{2, 3, 4\}$ are shown in Fig. 10 when the wake has settled after the step change. The comparison with the ADM supports the qualitative correspondence under steady-state conditions that was found in comparison with WTA. The good quality of the fit follows expectation because the wind tunnel has uniform inflow and a low turbulence intensity.

More importantly, this experiment can provide insight into the wake dynamics for changes in yaw misalignment with a high temporal resolution. The rotor-averaged wind speed for a virtual rotor is evaluated based on the PIV snapshots and shown in Fig. 11, together with the realised yaw angle. The actual yaw signal is used to replicate the experiment with the three FVW models.

    The value of VAF and NMAE for the three model simulations are listed in Table 2. The ADM and ADMR perform to a
similar level of accuracy in this mid-wake region. They are marginally outperformed by the ALM, which is considerably more computationally expensive. These results support the findings from Sect. 5.1 that the inclusion of rotation may improve the

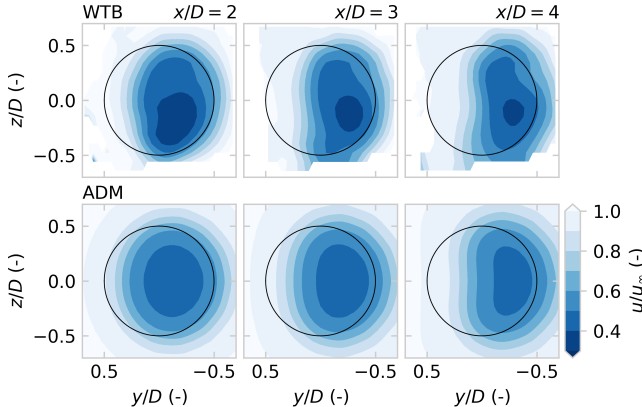

**Figure 10.** Flow slices showing streamwise velocity for yaw misalignment $\gamma = 20°$, comparing experimental measurements from WTB with ADM results.

**Table 2.** Fit quality of the time series of potential power estimates from replication of the WTB experiment, listing VAF (V) and NMAE (N). The experimental data and ADM estimate are illustrated in Fig. 11.

|     | ADM | | ADMR | | ALM | |
| --- | --- | --- | --- | --- | --- | --- |
| $x$ | V (%) | N (%) | V (%) | N (%) | V (%) | N (%) |
| 1 D | 90.8 | 4.3 | 90.8 | 4.4 | 92.5 | 4.4 |
| 2 D | 92.1 | 5.6 | 92.1 | 5.7 | 94.9 | 5.6 |
| 3 D | 91.5 | 6.3 | 91.6 | 6.5 | 94.5 | 4.9 |
| 4 D | 92.8 | 5.2 | 92.6 | 5.5 | 94.2 | 4.8 |

qualitative flow representation in the mid to far wake, but the added complexity does not appear necessary for control purposes.

The replication of the yaw step experiment with the ADM in Fig. 11 shows that the ADM estimates a potential power

improvement of similar magnitude as a result of the wake deflection by yaw misalignment. The low-frequency changes show the delays of control effects propagating downstream through the wake. These slow dynamics are also well-represented in the model estimate. However, some turbulence develops in the wake in the wind tunnel that causes variations in the velocity deficit which are not accounted for in the FVW. Considering yaw control is quite slow to actuate, it is more important that the slower dynamics are properly represented than the resolution of turbulence at smaller time scales.

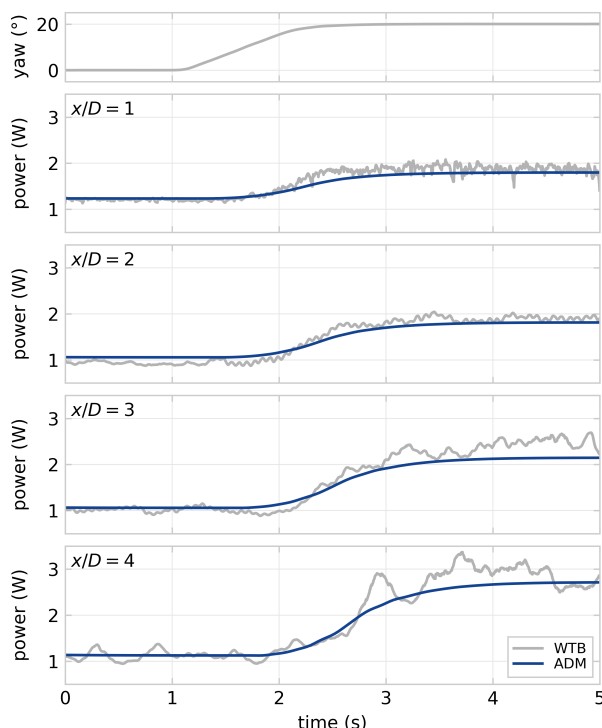

**Figure 11.** The realisation of the $\gamma = 20°$ step in yaw angle measured in WTB is shown in the top graph. The rotor-averaged potential power is calculated directly downstream of the turbine from for $x/D = \{1, 2, 3, 4\}$, comparing the experimental data and a simulation with the ADM. The FVW implementation of the ADM follows the dynamics of wake redirection well and estimates a potential power improvement of similar magnitude, although it lacks the turbulent disturbances.

## 5.3 Wind direction variation – WTC

The WTC dataset provides many performance measurements for varying yaw angle and wind direction. It is replicated only with the ADM, as it appears to perform similar to the ADMR and the computational expense of the ALM is prohibitive for control optimisation in the mid to far wake.

The replication of the experiment with BW30 controller settings with the ADM yields a power estimate for which the time series is shown in Fig. 12. The conversion from available aerodynamic power to generator power is performed with a first-order rotor model for which the inertia and controller settings are estimated based on the upstream turbine measurements. The downstream power estimate is then computed using the same turbine model. The turbine model fit has an inertia $J = 0.6\,\mathrm{kg\,m^2}$. The gains for the torque controller polynomial are estimated to be $k_1 = -3.7 \times 10^{-5}\,\mathrm{Nm\,rad\,s^{-1}}$ and $k_2 = 6.8 \times 10^{-6}\,\mathrm{Nm\,rad^2\,s^{-2}}$ for the upstream turbine and $k_1 = -2.1 \times 10^{-5}\,\mathrm{Nm\,rad\,s^{-1}}$ and $k_2 = 5.5 \times 10^{-6}\,\mathrm{Nm\,rad^2\,s^{-2}}$ for the downstream turbine. The estimated efficiency in converting potential aerodynamic power to generator power is $\eta = 54\,\%$.



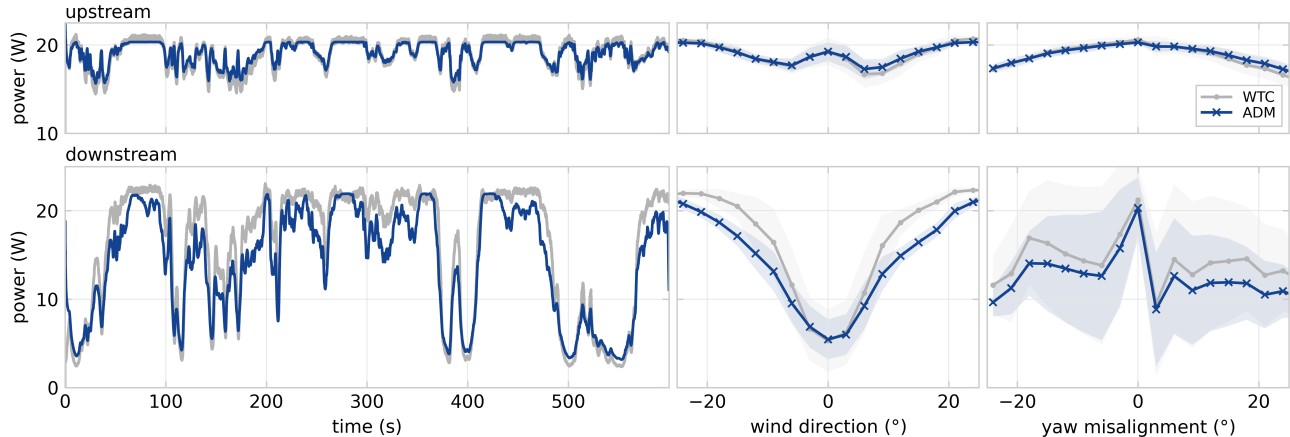

**Figure 12.** Power predictions with the ADM are compared with experimental data from WTC for the BW30 control setting. The time series fit VAF is $92.1\%$ for the upstream turbine and $93.3\%$ for the downstream turbine. The power curves for wind direction and yaw misalignment show mean and standard deviation in $3°$ bins. The power variation of the upstream turbine is completely due to variation in yaw misalignment from the open-loop yaw controller. The downstream power is affected both by yaw control on the upstream turbine and translation for modelling wind direction variation. For larger wind direction magnitudes, the downstream turbine is no longer in waked conditions. The downstream effects of yaw misalignment are clouded by the variation of wind direction and therefore, the power-yaw curves filtered for alignment with the wind direction are presented in Figure 13.

The upstream turbine power is estimated with a NMAE of $1.7\%$ and a VAF of $92.1\%$. However, the downstream turbine power estimate is primarily relevant for evaluation of the performance of the wake model. The downstream power series estimate achieves a NMAE of $8.8\%$ and VAF of $93.3\%$, which indicates most of the wake dynamics are accounted for. A similar fit quality is achieved for the power estimates with the other control settings from (Hulsman et al., 2023), which are

listed in Table 3. The first-order rotor model appears adequate in accounting for the delays due to rotor dynamics and the efficiency term captures most of the losses in power conversion.

Figure 12 also presents a statistical analysis of these power signals, where the mean and standard deviation are illustrated based on $3°$ wind direction bins. As expected from the time series, there is a close fit for the upstream power estimate. The variation in power over wind direction is due to changes in yaw angle based on the control strategy. The downstream power

estimate matches well for wind directions close to $0°$. For wind directions away from $0°$, where the downstream turbine is translated out of the waked conditions, the FVW underpredicts the power production on the downstream turbine. It is noteworthy that, in the wind tunnel, the downstream power exceeds the upstream power generated by about $6\%$ for wind directions where there is no aerodynamic wake interaction.

Besides possible differences between the model turbines themselves, this increase in power is likely due to a combination

of two effects. First, the inflow has a shear profile with a power-law exponent of $0.28$ and the downstream turbine has a higher hub height by $0.16\,D$. Consequently, it experiences a higher rotor-average velocity in unwaked conditions. This shear layer is




**Table 3.** Fit quality of the time series of generator power estimates from replication of the WTC experiment with the ADM, listing VAF (V) and NMAE (N). The experimental data and power estimate for the BW30 control strategy are illustrated in Fig. 12.

|         | upstream |         | downstream |         |
|---------|----------|---------|------------|---------|
| control | V (%)    | N (%)   | V (%)      | N (%)   |
| BW600   | 92.7     | 3.7     | 95.2       | 7.6     |
| BW300   | 92.5     | 1.7     | 93.5       | 8.7     |
| BW120   | 93.0     | 1.5     | 93.7       | 8.7     |
| BW60    | 92.6     | 1.7     | 92.9       | 9.1     |
| BW30    | 92.1     | 1.7     | 93.3       | 8.8     |

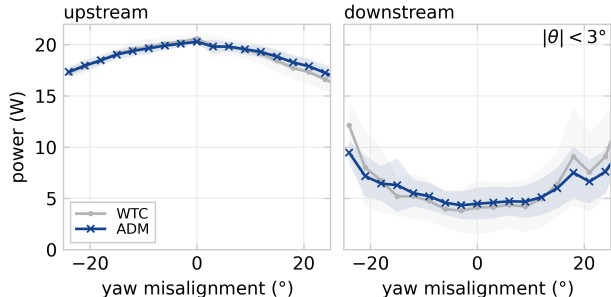

**Figure 13.** Wind turbine power curve for the modelled yaw misalignment distribution, showing mean and standard deviation of power in $3°$ yaw angle bins for the power data shown in Fig. 12. Data has been filtered to only consider those sections where the turbines are aligned with the wind direction, i.e. $|\theta| < 3°$. The wind tunnel power-yaw curve of the upstream turbine is slightly asymmetric due to sheared inflow. The predicted downstream power matches quite well for these aligned conditions.

unaccounted for in the FVW which assumes uniform inflow. Second, the wind tunnel has a limited $3\,\mathrm{m} \times 3\,\mathrm{m}$ cross section, which means there are blockage and speed-up effects due to the presence of the upstream turbine. These effects are especially amplified when the modelled wind direction $|\theta| > 20°$. The lateral flow component due to translation of the downstream turbine
is neglected.

The impact of wake steering through yaw misalignment on downstream power production is clouded in Fig. 12 because of the variation in wind direction. In order to analyse the model representation of wakes under yaw misalignment, the power signals are filtered for wind directions with fully waked conditions, i.e. $-3° < \theta < 3°$. The mean and standard deviation of power is calculated for $3°$ yaw angle bins and shown in Fig. 13.

The power-yaw curve of the upstream turbine is slightly asymmetric in the wind tunnel experiment due to the operation under sheared inflow conditions. This asymmetric power profile is not represented in the ADM because of the model symmetry and



uniform inflow. The downstream expected power production matches well given the model simplicity. The benefit of wake redirection is slightly underestimated for large misalignment angles.

The important aspect of the ADM estimate of power for these two turbines is that the trends are captured well. In the end, what matters for control optimisation for wake steering is accuracy in representing the optimal operating point more so than exactness in the predicted power. Additional error integration and state estimation could always be included if so required, such as for tracking a power reference.

## 6 Conclusions

Three free-vortex wake wind turbine models (ADM, ADMR, ALM) are presented in this work for the prediction of wake dynamics under yaw misalignment for control optimisation. The two highlights in this work are (i) a study of parameter sensitivity and convergence and (ii) a comparison with three sets of experimental data from wind tunnel measurements in order to validate the power predictions in the far wake.

The parameter and convergence study indicated that the best results for mid- to far-wake predictions are achieved with ADM, i.e. when the wake model has minimum complexity. The addition of rotation does improve qualitative agreement of flow fields with experimental data, but does not necessarily improve power predictions under yaw misalignment. The ALM may generate a slightly more accurate response, but the computational cost is prohibitive for use in online control optimisation for wake redirection.

The comparison with experimental data shows that the FVW models can provide meaningful predictions for power optimisation, even under the assumption of uniform inflow. There is a strong agreement with experimental data in terms of steady-state wake deflection, dynamic response to yaw change, and power estimates with yaw control and wind direction variation.

In conclusion, the ADM appears suitable for control optimisation for wake redirection considering mid- to far-wake effects, a range from approximately $1\,D$ to $7\,D$. As such, it could play a central role in the development of novel model-based strategies for wind farm flow control. These new controllers could further improve wind farm energy yield as more accurate wake dynamics are included in control optimisation for wake redirection.

For near-wake stability and rotor-plane effects, the ALM has added value as it models individual blades and tip vortices. Further downstream, large wake deformations under yaw misalignment limit the usefulness of the vortex filament approach. A transition to vortex particles or engineering wake models may be a suitable option to continue wake predictions further downstream.

*Code and data availability.* Model code supporting the work in this paper is available at 10.4121/e32a9868-c5ea-43d3-8969-b1908662b2b2 (van den Broek, 2023). Experimental data from WTA is available at DOI 10.5281/zenodo.5734877 (Hulsman et al., 2021), from WTB at DOI 10.4121/70ae7f4c-f87f-45f1-8360-f4958a60249f.v1 (van der Hoek et al., 2023b), and from WTC will be made available with (Hulsman et al., 2023).



*Author contributions.* Maarten J. van den Broek: conceptualisation, methodology, software, validation, investigation, writing – original daft, visualisation. Delphine De Tavernier: conceptualisation, writing – review & editing, Paul Hulsman: investigation – experiments WTA&WTC, writing – review & editing, Daan van der Hoek: investigation – experiment WTB, writing – review & editing, Benjamin Sanderse: writing – review & editing, supervision. Jan-Willem van Wingerden: writing – review & editing, conceptualisation, resources, funding acquisition.

*Competing interests.* At least one of the (co-)authors is a member of the editorial board of Wind Energy Science.

*Acknowledgements.* This work is part of the research programme "Robust closed-loop wake steering for large densely spaced wind farms" with project number 17512, which is (partly) financed by the Dutch Research Council (NWO). The experimental data (labelled WTA and WTC) is acquired in the scope of the research project "CompactWind II" (Ref. Nr. 0325492H) and funded by the Federal Ministry for Economic Affairs and Energy according to a resolution by the German Federal Parliament.



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
