# Peer review of "Free-vortex models for wind turbine wakes under yaw misalignment – a validation study on far-wake effects"

_Wind Energy Science, 2023_

## Referee Comment (RC2)

**General Comments:**

The article by van den Broek et al. uses a free-vortex model with three different wind turbine representations, 2 different actuator discs and one "actuator line", to study wind turbine wakes behind yawing turbines. First, a numerical sensitivity study is performed followed by a comparison against three wind tunnel data sets with increasing complexity and dynamics.

The written is generally fine and the article holds interesting aspects, although they are mainly related to the dynamical experiments from other studies, where one of the three studies is currently unpublished. However, the main problem is that the validation is misleading to a certain degree, as a number of important phenomena are neglected. Additionally, there are issues with the associated accuracy/error quantification, which is not up to state-of-the-art in terms of validation practices in general and for vortex methods in particular, where the proposed model lacks novelty. Therefore, I unfortunately can not recommend the article for publication.

**Specific Comments:**

**1. Validation.**
Validation studies are important to evaluate and continue to improve numerical models and build confidence and the scientific foundation for any subsequent studies. So to some degree it is very difficult to achieve a "conclusive" validation (line 11), as there will be need of more detailed validations in the future, even if this study was published. But more importantly, I don't think the present study fully qualifies as a validation study for several reasons.

a. The modeled physics are not the same as in the measurements, and the authors neglects the impact rather casually. There are three main aspects of lacking physics:
    i. no shear modeling
    ii. uniform loaded turbines (it is unclear if the experimental turbines have uniform loading distributions, but most wind tunnel turbine models will not)
    iii. no turbulence modeling
    All, but particular the last two, are known to be very important for the wake development. No turbulence is not equivalent to 1% turbulence intensity and the authors does not address the limitations related to their uniform load modeling. Below are several references, which address previous work on both uniform loading and other tools based on vortex methods, which captures the three aspects.
b. The numerical verification/sensitivity study in 3.1-3.4 is generally fine, and important to test. But I am not convinced that these tests are sufficient to prove that the methods are good enough to solve the dynamic yaw problem, nor that the chosen metrics are representative, when the different model fidelities clearly affect the

physics. For instance, the wake expansion is presumably different and therefore the rotor averaged deficit (Equation (20)) could be misleading and potentially cause the deviation seen in Figure 9.

c. The balance of computational and physical considerations seem off. Computational costs are important, but so are the physics. There are at least 5 instances in the article (line 192-195, 211, 228, 370, 407), where simplifying choices are made purely based on computational efficiency without any consideration of the physical impact. It does not provide a fair comparison to first make choices based only on computational efficiency, and then analyse/conclude that the method most sensitive to these numerical choices (ALM) is performing poorly (line 374-376 and 380-384). The conclusion that ADM provides the best balance of computational cost and accuracy therefore appears selective as the choices inevitably lead to those conclusions. Furthermore, this balance is not quantified, and the article would benefit from reporting the computational cost of the three different turbine representations. Before, seeing the computational cost, I would still question if the recommended method is actually fast enough for dynamic wind farm control applications. Finally, if computational speed is the only criteria, why does the authors not simply use the well-known FLORIS model (https://github.com/NREL/floris)?

**2. Accuracy and acceptable errors.**
The article does not sufficiently deal with and analyze the required accuracy. For example, line 398 states that added complexity is not necessary for control purposes. I would argue that the wake behind a yawed turbine is more complex than the wake behind a non-yawed turbine, so the modeling choices should reflect that, see Boorsma et al. reference below. Again, if the complexity is not important, then why not use FLORIS? I suggest the authors to compare the VFW results with FLORIS, which have previously been shown to provide similar accuracy at low costs, particular for mid- and far wakes. The improvements of power (Figure 11) are also only assessed qualitatively to be of "similar magnitude". How does this compare to the quantified VAF and NMAE? The NMAE and VAF does indicate qualitatively that most of the dynamics are accounted for (line 418), but the remaining 6-9% could cause the consistent under-prediction of ADM seen in Figure 12, which is presumably related to lack of turbulence.

Finally and most importantly, I think the authors should reflect more on what accuracy is required and hence what are acceptable errors for wind farm control studies. Many previous studies on wind farm control have suffered from large/unknown errors and uncertainty, so I think the authors need to address if errors of 6-9% are acceptable relative to the potential power increase seen in wind farm control studies. Therefore, it is also too casual to state that the accuracy in operating point is more important than exactness in predicted power (line 445-446). How does the authors intend to determine "optimal operating point" with methods that give inaccurate power predictions? The understanding and reduction of errors and uncertainties are critical to make wind farm flow control successful and applicable on industrial scale, as discussed in the large international benchmark reported here:

- Göçmen, T., Campagnolo, F., Duc, T., Eguinoa, I., Andersen, S. J., Petrović, V., Imširović, L., Braunbehrens, R., Liew, J., Baungaard, M., van der Laan, M. P., Qian, G., Aparicio-Sanchez, M., González-Lope, R., Dighe, V. V., Becker, M., van den Broek, M. J., van Wingerden, J.-W., Stock, A., Cole, M., Ruisi, R., Bossanyi, E., Requate, N., Strnad, S., Schmidt, J., Vollmer, L., Sood, I., and Meyers, J.: FarmConners wind farm flow control benchmark – Part 1: Blind test results, Wind Energ. Sci., 7, 1791–1825, https://doi.org/10.5194/wes-7-1791-2022, 2022.

To be honest, I would not trust future results on wind farm control based on the presented validation of the VFW tool. Therefore, I think the aim of validating a simple model to use for wind farm control studies is counterproductive from a scientific (and societal) perspective, where we need more accuracy and confidence in results, not just speed.

**3. Lack of novelty and references.**
Vortex methods have a lot of potential, and a number of tools based on current state-of-the-art vortex methods exists, which have been validated in more detail than the present work. The novelty of the present work is therefore mainly related to the dynamic experimental results published elsewhere, not the model. Several of these tools include turbulence modeling, and should be properly referenced. A couple of recent publications are provided here, but I suggest to search more on these tools and current state-of-the-art vortex methods:
- Bergua, R., Robertson, A., Jonkman, J., Branlard, E., Fontanella, A., Belloli, M., Schito, P., Zasso, A., Persico, G., Sanvito, A., Amet, E., Brun, C., Campaña-Alonso, G., Martín-San-Román, R., Cai, R., Cai, J., Qian, Q., Maoshi, W., Beardsell, A., Pirrung, G., Ramos-García, N., Shi, W., Fu, J., Corniglion, R., Lovera, A., Galván, J., Nygaard, T. A., dos Santos, C. R., Gilbert, P., Joulin, P.-A., Blondel, F., Frickel, E., Chen, P., Hu, Z., Boisard, R., Yilmazlar, K., Croce, A., Harnois, V., Zhang, L., Li, Y., Aristondo, A., Mendikoa Alonso, I., Mancini, S., Boorsma, K., Savenije, F., Marten, D., Soto-Valle, R., Schulz, C. W., Netzband, S., Bianchini, A., Papi, F., Cioni, S., Trubat, P., Alarcon, D., Molins, C., Cormier, M., Brüker, K., Lutz, T., Xiao, Q., Deng, Z., Haudin, F., and Goveas, A.: OC6 project Phase III: validation of the aerodynamic loading on a wind turbine rotor undergoing large motion caused by a floating support structure, Wind Energ. Sci., 8, 465–485, https://doi.org/10.5194/wes-8-465-2023, 2023.
- S. Perez-Becker, J. Saverin, R. Behrens de Luna, C. Combreau, M.-L. Ducasse, D. Marten, A. Bianchini, 2022, "D2.2. Validation Report of QBlade-Ocean", FLOATECH Deliverable, Technical Report
- Ramos-García et al. 2023, "Multi-fidelity vortex simulations of rotor flows: Validation against detailed wake measurements", https://www.sciencedirect.com/science/article/pii/S0045793023000154
- Alvarex, E. J. and Ning, A., 2022, "Reviving the Vortex Particle Method: A Stable Formulation for Meshless Large Eddy Simulation", https://doi.org/10.48550/arXiv.2206.03658

A similar study have also been performed against wind tunnel data using LES, where the uniform and non-uniformly loaded ADM were compared, and found that uniformly loaded ADM is not sufficient to capture the correct physics. The authors should reflect on such results:

- Lin, M. and Porté-Agel, F.: Large-eddy simulation of a wind-turbine array subjected to active yaw control, Wind Energ. Sci., 7, 2215–2230, https://doi.org/10.5194/wes-7-2215-2022, 2022.

Other studies show that even actuator lines are not necessarily accurate enough for turbines operating in high yaw with the increased complexity of full scale experiments, and this is particular related to the simplifying assumption of using 2D airfoils, see for instance the following reference:

- Boorsma, K., Schepers, G., Aagard Madsen, H., Pirrung, G., Sørensen, N., Bangga, G., Imiela, M., Grinderslev, C., Meyer Forsting, A., Shen, W. Z., Croce, A., Cacciola, S., Schaffarczyk, A. P., Lobo, B., Blondel, F., Gilbert, P., Boisard, R., Höning, L., Greco, L., Testa, C., Branlard, E., Jonkman, J., and Vijayakumar, G.: Progress in the validation of rotor aerodynamic codes using field data, Wind Energ. Sci., 8, 211–230, https://doi.org/10.5194/wes-8-211-2023, 2023.

**Technical Corrections:**

**1. Unpublished third study**
The third experimental study in 4.1.3 is not yet submitted for publication so it is hard to understand the experimental setup. Perhaps it is the lack of drawing showing the experimental setup, but it is hard to understand how moving the second turbine laterally out of the wake of the first turbine should correspond to a change of wind direction?

**2. Language and minor corrections**
A number of wordings are in my opinion misleading. Here are a number of major ones, but the article would improve by general clarification of specific comments.
   a. The FVW is in my opinion not a surrogate (line 5 + line 59). A surrogate is typically data-driven model based purely on regression of input-output without physical modelling. The FVW does solve simplified physical equations. Have a look at: https://en.wikipedia.org/wiki/Surrogate_model
   b. The three different models (ADM, ADMR, ALM) are representations of the turbine, not wake models (line 73). The FVW models the wake based on the forcing from the different turbine representations.
   c. "Actuator line" is the method used to represent individual blades in CFD (for instance LES) where forces are smeared numerically, while the term used in vortex methods is typically "lifting line", see your own reference van Kuik, 2018.
   d. The wake transitions (line 231) does not seem "natural", but as a purely numerical artifact.

e. The trends of the model results might be meaningful, but it does not mean that they are accurate (line 458).

f. Equation (12) does not show pressure jump, but vorticity. They are of course related as stated in 136-137, but rephrase to match the equation.

g. line 161: To my understanding, the bound vorticity of the root vortex should be opposite to the tip vortices.

h. Line 342: If you have an induction factor of a = 0.33, then CT = 0.89 with same number of significant digits.

---

## Author Comment (AC1)

**Response to Reviewer Comments**

First of all, we thank the reviewers for taking the time to read our manuscript and providing their constructive feedback. This document provides our response and describes the changes that have been made in the light of your thoughtful comments. In the subsequent sections, we will address the review reports from each of the reviewers individually.

**1 Reviewer 1**

**1.1 General comments**

> The paper provides an extensive analysis of the use of the free-vortex model for wind turbines in yaw using the actuator disk, actuator disk with rotation, and actuator line formulations. The results demonstrate that these models perform well at reconstructing the flow and are theoretically sound. The paper is generally clear and well written. However, some improvements to the paper could improve its usefulness to readers.

We thank the reviewer for their positive comments and suggestions for improvements!

> My only significant critique is that it is difficult to follow the equations to compare to other theoretical formulations and papers. More details about the coordinate systems and careful definitions of variables would go a long way in helping readers replicate the results. The yawed wind turbine literature is filled with imprecisely defined variables like induction factor that are not consistent across papers. This makes it very hard to replicate studies and results and has resulted in citation errors. A careful writing would prevent this paper from having that flaw.

We value the clarity of the presented work and have addressed the precision of definitions. Details of the adjustments we have made are addressed in the response to the specific comments.

> I also suggest that more details about the computational cost would help readers understand the best application of this approach. Comparisons to LES and engineering models, for example, could help users weight the costs and benefits. Scaling with number of turbines, domain size, and resolution would also be helpful.

In response to the reviewer suggestion to include more details on computational cost, we have included some benchmark numbers for the models presented in the paper.

**1.2 Specific comments**

> Section 2.2.1: Could you elaborate on the use of the induction factor. Is $a$ measured as streamwise induction or induction normal to the rotor? For ADM-R and ALM, how does $a$ vary with radius?

The induction factor refers to the axial induction normal to the rotor as defined in momentum theory. It is used in this work as a control input to define thrust and power coefficients. Therefore, it is not measured from flow velocity and the actual velocity deficit may differ from what the induction factor suggests. The description in paragraph 2.2.1 has been updated to clarify. The three models assume uniformly loaded rotor discs and, therefore, no radial variation of induction factor is applied.

> Section 2.2.2: A few comments on this section:
> - The ADM vortex system should be a semi-infinite cylinder. How does the stream-

> wise and azimuthal discretization relate the the streamwise velocity? Does this include induction? Please relate $\Delta t$ to the streamwise velocity.

In the current model formulation, the vortex cylinder has finite length. A note on this has been added in Section 2. The time step $\Delta t$ determines the streamwise discretisation. It is related to the undisturbed streamwise velocity $u_\infty$ and rotor diameter $D$ as noted in Section 3, where the default time step is chosen $\Delta t \cdot u_\infty / D = 0.3$. A reference to the parameters listed in Table 1 has been added to Section 2.2.2.

> - In what direction is the vorticity pointing? Is it purely azimuthal vorticity? A yawed wind turbine should also generate streamwise vorticity, which can be predicted analysitically. Does your equation provide this vorticity?

The vorticity generated by the ADM is purely azimuthal as long as the turbine is yaw-aligned with the free-stream wind direction. The deformation of vortex rings from a rotor under yaw misalignment provides the streamwise vorticity. This has been noted in Section 2.2.2.

> - Sections 2.2.3 and 2.2.4: I have similar questions in these sections to Section 2.2.2. It is unclear exactly what the coordinate system is, in which direction the vorticity is pointing, and how the system is discretized relative to the streamwise velocity.

The direction of the vortex circulation is now illustrated in Fig. 1. The discretisation in the streamwise direction is discussed in Section 3.1, which notes that the streamwise spacing between vortex filaments is related to the simulation time step that scales with the free-stream velocity and rotor diameter.

> Figure 2: How is vortex filament spacing normalized? Is it by delta t? Please put on the axis label. Is the velocity error dimensionless? Equation 18 would have units of velocity. Please note the units in the ordinate label.

We thank the reviewer for noticing the omission of normalisation. We have adjusted the axis labels in Figures 2-4 to include normalisation by rotor diameter and inflow velocity to be consistent with the rest of the manuscript. This clarifies the calculations as they were implemented.

> Section 3: A few comments on Section 3:
> - You could also compare the results to theory since the flow behind an unyawed turbine is well known.

A comparison with a theoretical result for a yaw-aligned rotor could indeed be included. However, our main interest lies with the consistency and convergence of the current numerical methods.

> - Please also report results for yawed wind turbines. The unyawed case is much more difficult to reconstruct and errors in the modeling of yaw would not be apparent without this comparison.

The results on convergence of time discretisation for a yaw-misaligned turbine show similar trends to those for a yaw-aligned turbine. We agree that the results for the yawed turbine are relevant to the work and have therefore replaced Figure 3 with the convergence results for the wake of a yaw-misaligned rotor. The results for a yaw-aligned rotor have now been omitted for conciseness, but are included in the supplemental data. The azimuthal spatial discretisation in Section 3.3 already considers a yaw-misaligned turbine. Furthermore, errors in the modelling of yaw – if present – would also appear in the validation against experimental data in Section 5.

> - Were these results all using the steady state experiments from Section 4?

The results in Section 3 concern numerical convergence and we have now stated this more explicitly in the introduction to Section 3. The parameters used for the reference solutions are indicated in the text and figure captions.

> - Can you provide an estimate of the computational cost with this scaling and how it compares to other methods (e.g. LES or engineering models).

As noted for the general comment, we now include some benchmark numbers comparing computational cost of the three models at the end of Section 3.1. We place the benchmark into perspective of evaluating a steady-state wake approximation using an engineering wake model or simulating the dynamic wake in large-eddy simulation.

> Section 4.1: Some more repetition of the basics of these experiments here would be useful so the reader does not have to go back to these other papers.

We have provided some additional details on the wind tunnels and measurement set-ups for the experimental studies in Section 4.1.

> Section 4.1.1 and 4.1.2: Paragraphs typically need 3-4 sentences. I suggest you remove the line breaks and make these entire sections into one paragraph.

We have adjusted these sections by removing several line breaks.

**2 Reviewer 2**

**2.1 General comments**

> The article by van den Broek et al. uses a free-vortex model with three different wind turbine representations, 2 different actuator discs and one "actuator line", to study wind turbine wakes behind yawing turbines. First, a numerical sensitivity study is performed followed by a comparison against three wind tunnel data sets with increasing complexity and dynamics. The written is generally fine and the article holds interesting aspects, although they are mainly related to the dynamical experiments from other studies, where one of the three studies is currently unpublished. However, the main problem is that the validation is misleading to a certain degree, as a number of important phenomena are neglected.
>
> Additionally, there are issues with the associated accuracy/error quantification, which is not up to state-of-the-art in terms of validation practices in general and for vortex methods in particular, where the proposed model lacks novelty. Therefore, I unfortunately can not recommend the article for publication.

We thank the reviewer for his critical comments which we have used to improve our manuscript. Firstly, the models are presented in light of optimisation for yaw control, with the aim of achieving real-time model-predictive control for wake steering. Several important phenomena are neglected on purpose, which is precisely why model validation is essential. We, as authors, take it seriously that the reviewer feels the validation is misleading. We hope our response, and the associated adjustments to the manuscript, can clarify our intent with the study. We further address the specific comments below.

**2.2 Specific comments**

**2.2.1 Validation**

> Validation studies are important to evaluate and continue to improve numerical models and build confidence and the scientific foundation for any subsequent studies. So to some degree it is very difficult to achieve a "conclusive" validation (line 11), as there will be need of more detailed validations in the future, even if this study was published. But more importantly, I don't think the present study fully qualifies as a validation study for several reasons.

Thank you for pointing out the importance of validation studies for numerical models, on that point we agree. Describing the current validation as "conclusive" may have been lightly overstated and, to soften the claim, we have removed the descriptor. We do believe the current study qualifies as a validation study for the presented purpose and describe so in response to your further remarks.

> a. The modeled physics are not the same as in the measurements, and the authors neglects the impact rather casually. There are three main aspects of lacking physics:
>   i. no shear modeling
>   ii. uniform loaded turbines (it is unclear if the experimental turbines have uniform loading distributions, but most wind tunnel turbine models will not)
>   iii. no turbulence modeling
>
> All, but particular the last two, are known to be very important for the wake development. No turbulence is not equivalent to 1% turbulence intensity and the authors does not address the limitations related to their uniform load modeling. Below are several references, which address previous work on both uniform loading and other tools based on vortex methods, which captures the three aspects.

We agree that the modelled physics are not the same as in the experiments. This is unavoidable, since any model is a simplification of reality. Here the question is to which degree the modelled physics are accurate enough in the sense that the model can be used for control optimisation. The corresponding assumptions for simplifying the physics in the models are made explicitly and are part of the model formulations in Section 2 of the manuscript. These are also clearly described as differing from the conditions in the wind tunnel experiments. It is because of the simplifying assumptions that this validation study is necessary to evaluate suitability to the intended purpose. We make no claim that the models precisely replicate the experimental conditions and describe in the results and discussion that these assumptions may limit the accuracy of the current model.

> b. The numerical verification/sensitivity study in 3.1-3.4 is generally fine, and important to test. But I am not convinced that these tests are sufficient to prove that the methods are good enough to solve the dynamic yaw problem, nor that the chosen metrics are representative, when the different model fidelities clearly affect the physics. For instance, the wake expansion is presumably different and therefore the rotor averaged deficit (Equation (20)) could be misleading and potentially cause the deviation seen in Figure 9.

The performance in terms of modelling the dynamic yaw problem is extensively treated in the comparison with experimental data. The rotor-averaged deficit is chosen because a downstream turbine inherently averages flow quantities over the rotor-area in the power output. A sentence explaining this motivation has been added to Section 3.3. Note that the wake expansion is consistent between the three models as illustrated in Figure 8.

> c. The balance of computational and physical considerations seem off. Computational costs are important, but so are the physics. There are at least 5 instances in the article (line 192-195, 211, 228, 370, 407), where simplifying choices are made purely based on computational efficiency without any consideration of the physical impact. It does not provide a fair comparison to first make choices based only on computational efficiency, and then analyse/conclude that the method most sensitive to these numerical choices (ALM) is performing poorly (line 374-376 and 380-384). The conclusion that ADM provides the best balance of computational cost and accuracy therefore appears selective as the choices inevitably lead to those conclusions. Furthermore, this balance is not quantified, and the article would benefit from reporting the computational cost of the three different turbine representations. Before, seeing the computational cost, I would still question if the recommended method is actually fast enough for dynamic wind farm control applications. Finally, if computational speed is the only criteria, why does the authors not simply use the well-known FLORIS model (https://github.com/NREL/floris)?

We agree that a quantification of the computational cost is beneficial to the manuscript and therefore have included a benchmark timing for each of the models at the end of section 3.1, as mentioned in the response to reviewer 1. This highlights that the ADM provides sufficient speed for real-time control optimisation. Of course, computational speed is not the only criterium, which is why the convergence study is included to quantify the numerical errors and the different models are evaluated with respect to the wind tunnel data.

We have now also included a comparison with steady-state results in Section 5.2 to illustrate the difference between steady-state wake predictions and simulations with dynamics. These results demonstrate that the time-resolved model predicts the transients in wake development that are not represented in steady wake models.

The well-known FLORIS model is also a steady-state approximation of the dynamics of the

wind turbine wake and therefore unable to predict the transients of changes propagating through the wake. We aim to make the step towards dynamic model-predictive control for wake steering. An efficient, reasonably accurate, time-resolved model of the wind turbine wake is, in our view, at the core of such a control strategy because it enables incorporation of these transient dynamics.

**2.2.2 Accuracy and acceptable errors**

> The article does not sufficiently deal with and analyze the required accuracy. For example, line 398 states that added complexity is not necessary for control purposes. I would argue that the wake behind a yawed turbine is more complex than the wake behind a non-yawed turbine, so the modeling choices should reflect that, see Boorsma et al. reference below. Again, if the complexity is not important, then why not use FLORIS? I suggest the authors to compare the VFW results with FLORIS, which have previously been shown to provide similar accuracy at low costs, particular for mid- and far wakes.

From the results presented, the additional complexity of the rotating ADM and the lifting-line model does not contribute to large improvements in accuracy compared to the regular ADM when considering the power predictions.

We agree, as you would argue, that the wake of a yaw-misaligned turbine is more complex than that of a yaw-aligned turbine. For that reason, the rings of vorticity shed by the actuator-disc are constructed from individual straight-line vortex filaments. These facilitate the deformation of the vortex rings and formation of the characteristic curled wake, as has now been noted in Section 2.2. This modelling choice thus already reflects the complexity of the wake from a turbine under yaw misalignment – circular vortex rings would have been sufficient for a yaw-aligned turbine with a symmetric wake.

As noted above, FLORIS is a steady-state engineering model of the wake. Under steady conditions, it would not make sense to develop a time-resolved model to perform wake steering. However, with the aim of performing dynamic model-predictive control for wake steering under time-varying conditions we believe a time-resolved model is necessary. We have added a comparison between dynamic and steady-state results in Section 5.2 that illustrates the limitations of steady-state assumptions.

> The improvements of power (Figure 11) are also only assessed qualitatively to be of "similar magnitude". How does this compare to the quantified VAF and NMAE? The NMAE and VAF does indicate qualitatively that most of the dynamics are accounted for (line 418), but the remaining 6-9% could cause the consistent under-prediction of ADM seen in Figure 12, which is presumably related to lack of turbulence. Finally and most importantly, I think the authors should reflect more on what accuracy is required and hence what are acceptable errors for wind farm control studies. Many previous studies on wind farm control have suffered from large/unknown errors and uncertainty, so I think the authors need to address if errors of 6-9% are acceptable relative to the potential power increase seen in wind farm control studies. Therefore, it is also too casual to state that the accuracy in operating point is more important than exactness in predicted power (line 445-446). How does the authors intend to determine "optimal operating point" with methods that give inaccurate power predictions? The understanding and reduction of errors and uncertainties are critical to make wind farm flow control successful and applicable on industrial scale, as discussed in the large international benchmark reported here:
> - Göçmen et al. (2022)

We expect that the current errors are acceptable, but cannot provide further proof in this study

than the evidence in the manuscript. We have rephrased throughout the manuscript to address that the current data – in our view – makes the ADM a suitable candidate for use in a wind farm flow control strategy, but that the current data does not prove that it is sufficiently accurate. Future work will then need to demonstrate whether the provided accuracy is sufficient with the effectiveness of a control strategy.

For the purpose of power maximisation, the optimised control signal will yield optimal power production if the optimal operating point of the model corresponds to that of the real plant, even though the predicted power may differ from reality. The results from the comparison with experiments indicate that there is a considerable level of correspondence in the power estimates and their dependency on yaw misalignment. As noted in the manuscript, further tuning and re-fining of the accuracy of power predictions could be included in a closed-loop control framework, which is why we expect the current error values to be acceptable for controller synthesis. The online adaptation of model parameters and state estimation by incorporating measurements has already been demonstrated to be effective in wind farm flow control (Doekemeijer et al., 2020).

> To be honest, I would not trust future results on wind farm control based on the presented validation of the VFW tool. Therefore, I think the aim of validating a simple model to use for wind farm control studies is counterproductive from a scientific (and societal) perspective, where we need more accuracy and confidence in results, not just speed.

The model validation is performed in light of control optimisation for wake steering. A simple, fast model with sufficient accuracy to provide meaningful power predictions is essential for that purpose. As noted in earlier answers, we can not yet state what level of accuracy is sufficient and have adjusted phrasing in our manuscript accordingly. Of course, the presented FVW model is not, and is not claimed to be, a suitable tool for validating control strategies. It is, and is presented as, a tool for synthesising a model-based controller for wind farm flow control. The validation study performed here is related to that objective and we believe the models add value as a tool for control optimisation.

**2.2.3 Lack of novelty and references**

> Vortex methods have a lot of potential, and a number of tools based on current state-of-the-art vortex methods exists, which have been validated in more detail than the present work. The novelty of the present work is therefore mainly related to the dynamic experimental results published elsewhere, not the model. Several of these tools include turbulence modeling, and should be properly referenced. A couple of recent publications are provided here, but I suggest to search more on these tools and current state-of-the-art vortex methods:

We believe that the novelty of the present work is related to the evaluation of model complexity and validation for power predictions in the mid to far wake for optimisation of dynamic yaw control signals.

> - Bergua et al. (2023), Perez-Becker et al. (2022), Ramos-García et al. (2023), Alvarez and Ning (2022)

Both Bergua et al. (2023) and Perez-Becker et al. (2022) study rotor loads and aerodynamics on floating platforms and lack consideration of wake effects. Ramos-García et al. (2023) studies the simulation of tip vortex stability in the near wake in a two-bladed rotor wake in a water channel, providing groundwork for high-fidelity studies into tip-vortex stability in the near wake of yaw-aligned rotors. Alvarez and Ning (2022) presents a reformulated vortex-particle method for large eddy simulation, computationally orders of magnitude more expensive than the models presented in our work. These differ from the current work which focuses on the computationally efficient

dynamic modelling of the mid to far wake of wind turbines operating under yaw misalignment.

> - A similar study have also been performed against wind tunnel data using LES, where the uniform and non-uniformly loaded ADM were compared, and found that uniformly loaded ADM is not sufficient to capture the correct physics. The authors should reflect on such results: Lin and Porté-Agel (2022)

Lin and Porté-Agel (2022) presents an interesting comparison of rotor modelling paradigms in large-eddy simulation under steady conditions. We have included a note on the assumption of uniform loading and possibility for improved wake deflection modelling in Section 5.1.

> - Other studies show that even actuator lines are not necessarily accurate enough for turbines operating in high yaw with the increased complexity of full scale experiments, and this is particular related to the simplifying assumption of using 2D airfoils, see for instance the following reference: Boorsma et al. (2023)

Boorsma et al. (2023) does consider a rotor operating under yaw misalignment, but is concerned with rotor loads and aerodynamics. The study does not present any results on wake simulations, which are the focus of our current work. The models shown in our manuscript are likely not the best choice for rotor aerodynamics, as that is not the intent with which they were chosen.

**2.3 Technical Corrections**

**2.3.1 Unpublished third study**

> The third experimental study in 4.1.3 is not yet submitted for publication so it is hard to understand the experimental setup. Perhaps it is the lack of drawing showing the experimental setup, but it is hard to understand how moving the second turbine laterally out of the wake of the first turbine should correspond to a change of wind direction?

Unfortunately, there has been a delay in the publication process of the third experimental study. The translation of the downstream turbine corresponds to a rotation of the two-turbine wind farm layout in the flow. It is translated along an arc segment such that the distance to the upstream turbine is constant. Effectively, this layout rotation changes the wind direction through the wind farm, which achieves an effect similar to rotating a wind farm on a turntable such as used, for example, by Campagnolo et al. (2020). We have clarified the description of the experimental setup in Section 4.1.3.

**2.3.2 Language and minor corrections**

> A number of wordings are in my opinion misleading. Here are a number of major ones, but the article would improve by general clarification of specific comments.
> a. The FVW is in my opinion not a surrogate (line 5 + line 59). A surrogate is typically data-driven model based purely on regression of input-output without physical modelling. The FVW does solve simplified physical equations. Have a look at: https://en.wikipedia.org/wiki/Surrogate_model

The term "surrogate model" has previously been used in model-based wind farm control such as Doekemeijer et al. (2021). The model is a simplified representation of the actual plant for control optimisation and could, therefore, be named a surrogate model. However, to avoid confusion with data-driven methods, we adjust the description of the model.

> b. The three different models (ADM, ADMR, ALM) are representations of the turbine, not wake models (line 73). The FVW models the wake based on the forcing from the different turbine representations.

We agree and have adjusted the phrasing here and checked it throughout the manuscript.

> c. "Actuator line" is the method used to represent individual blades in CFD (for instance LES) where forces are smeared numerically, while the term used in vortex methods is typically "lifting line", see your own reference van Kuik, 2018.

We adjust the name of the third rotor model from "actuator-line model (ALM)" to "lifting-line model (LLM)" to be in agreement with the referenced literature.

> d. The wake transitions (line 231) does not seem "natural", but as a purely numerical artifact.

We have adjusted the choice of words to not refer to this transition as "natural".

> e. The trends of the model results might be meaningful, but it does not mean that they are accurate (line 458).

We have rephrased the statement in the conclusions to better reflect that the data provides a measure of accuracy, yet does not prove whether or not that accuracy is sufficient.

> f. Equation (12) does not show pressure jump, but vorticity. They are of course related as stated in 136-137, but rephrase to match the equation.

We have rephrased the wording to match the equation.

> g. line 161: To my understanding, the bound vorticity of the root vortex should be opposite to the tip vortices.

The vorticity of the root vortex is indeed opposite to that of the tip vortices. However, that is accounted for in the definition of the vortex filaments. To clarify, the illustration of the FVW models in Figure 1 has been extended.

> h. Line 342: If you have an induction factor of a = 0.33, then CT = 0.89 with same number of significant digits.

We have noted the thrust coefficient to two significant digits.

**References**

Alvarez, E. J. and Ning, A. (2022). Reviving the Vortex Particle Method: A Stable Formulation for Meshless Large Eddy Simulation. 6(December).

Bergua, R., Robertson, A., Jonkman, J., Branlard, E., Fontanella, A., Belloli, M., Schito, P., Zasso, A., Persico, G., Sanvito, A., Amet, E., Brun, C., Campaña-Alonso, G., Martín-San-Román, R., Cai, R., Cai, J., Qian, Q., Maoshi, W., Beardsell, A., Pirrung, G., Ramos-García, N., Shi, W., Fu, J., Corniglion, R., Lovera, A., Galván, J., Nygaard, T. A., Dos Santos, C. R., Gilbert, P., Joulin, P. A., Blondel, F., Frickel, E., Chen, P., Hu, Z., Boisard, R., Yilmazlar, K., Croce, A., Harnois, V., Zhang, L., Li, Y., Aristondo, A., Mendikoa Alonso, I., Mancini, S., Boorsma, K., Savenije, F., Marten, D., Soto-Valle, R., Schulz, C. W., Netzband, S., Bianchini, A., Papi, F., Cioni, S., Trubat, P., Alarcon, D., Molins, C., Cormier, M., Brüker, K., Lutz, T., Xiao, Q., Deng, Z., Haudin, F., and Goveas, A. (2023). OC6 project Phase III: validation of the aerodynamic loading on a wind turbine rotor undergoing large motion caused by a floating support structure. *Wind Energy Sci.*, 8(4):465–485.

Boorsma, K., Schepers, G., Aagard Madsen, H., Pirrung, G., Sørensen, N., Bangga, G., Imiela, M., Grinderslev, C., Meyer Forsting, A., Shen, W. Z., Croce, A., Cacciola, S., Schaffarczyk, A. P., Lobo, B., Blondel, F., Gilbert, P., Boisard, R., Höning, L., Greco, L., Testa, C., Branlard, E., Jonkman, J., and Vijayakumar, G. (2023). Progress in the validation of rotor aerodynamic codes using field data. *Wind Energy Sci.*, 8(2):211–230.

Campagnolo, F., Weber, R., Schreiber, J., and Bottasso, C. L. (2020). Wind tunnel testing of wake steering with dynamic wind direction changes. *Wind Energy Sci.*, 5(4):1273–1295.

Doekemeijer, B. M., Kern, S., Maturu, S., Kanev, S., Salbert, B., Schreiber, J., Campagnolo, F., Bottasso, C. L., Schuler, S., Wilts, F., Neumann, T., Potenza, G., Calabretta, F., Fioretti, F., and van Wingerden, J. W. (2021). Field experiment for open-loop yaw-based wake steering at a commercial onshore wind farm in Italy. *Wind Energy Sci.*, 6(1):159–176.

Doekemeijer, B. M., van der Hoek, D., and van Wingerden, J. W. (2020). Closed-loop model-based wind farm control using FLORIS under time-varying inflow conditions. *Renew. Energy*, 156:719–730.

Göçmen, T., Campagnolo, F., Duc, T., Eguinoa, I., Andersen, S. J., Petrović, V., Imširović, L., Braunbehrens, R., Liew, J., Baungaard, M., van der Laan, M. P., Qian, G., Aparicio-Sanchez, M., González-Lope, R., Dighe, V. V., Becker, M., van den Broek, M. J., Van Wingerden, J. W., Stock, A., Cole, M., Ruisi, R., Bossanyi, E., Requate, N., Strnad, S., Schmidt, J., Vollmer, L., Sood, I., and Meyers, J. (2022). FarmConners wind farm flow control benchmark – Part 1: Blind test results. *Wind Energy Sci.*, 7(5):1791–1825.

Lin, M. and Porté-Agel, F. (2022). Large-eddy simulation of a wind-turbine array subjected to active yaw control. *Wind Energy Sci.*, 7(6):2215–2230.

Perez-Becker, A. S., Saverin, J., Behrens, R., Luna, D. E., Papi, F., Combreau, C., Ducasse, M.-L., Marten, D., and Bianchini, A. (2022). FLOATECH D2.2. Validation Report of QBlade-Ocean. Technical Report August.

Ramos-García, N., Abraham, A., Leweke, T., and Sørensen, J. N. (2023). Multi-fidelity vortex simulations of rotor flows: Validation against detailed wake measurements. *Comput. Fluids*, 255(August 2022):105790.